# Mitigating analytical variability in fMRI with style transfer

**Elodie Germani**[1]                                    ELODIEGERMANI@GMAIL.COM
**Camille Maumet**[*1]                                  CAMILLE.MAUMET@INRIA.FR
**Elisa Fromont**[*2]                                   ELISA.FROMONT@IRISA.FR
[1] *Univ Rennes, Inria, CNRS, Inserm, Rennes, France*
[2] *Univ Rennes, IUF, Inria, CNRS, Rennes, France*

**Editors:** Accepted for publication at MIDL 2025

## Abstract

We propose a novel approach to facilitate the re-use of neuroimaging results by converting statistic maps across different functional MRI pipelines. We make the assumption that pipelines used to compute fMRI statistic maps can be considered as a style component and we propose to use generative models, among which, Generative Adversarial Networks (GAN) and Diffusion Models (DM) to harmonize statistic maps across different pipelines. We explore the performance of multiple GAN and DM frameworks for style transfer. We developed an auxiliary classifier that distinguishes statistic maps from different pipelines, allowing us to validate pipeline transfer, but also to extend traditional sampling techniques used in DM to improve the transition performance. Our experiments demonstrate that our proposed methods are successful: pipelines can indeed be transferred as a style component, providing an important source of data augmentation for future studies.

**Keywords:** style transfer, generative models, analytical variability, fMRI, data re-use

## Introduction

Over the last decades, understanding brain functions has taken an important place in many research fields. With the development of neuroimaging techniques such as task-based functional MRI (fMRI), researchers can now explore the brain activity of individuals who perform predefined tasks, and better understand the neural correlates of cognitive processes.

However, the "reproducibility crisis" raised concerns about the reliability of published findings, including in neuroimaging (Button et al., 2013; Poldrack et al., 2017; Botvinik-Nezer and Wager, 2023), and prompted the adoption of new research practices. Efforts have been made to increase sample sizes by acquiring data from a larger number of participants for a few numbers of cognitive tasks (*e.g.* UK Biobank (Sudlow et al., 2015)) or a small number of participants on a larger number of cognitive tasks (*e.g.* Individual Brain Charting (Pinho et al., 2018)). However, the number of research questions that can be explored is always limited by the characteristics of each dataset.

With the increased adoption of data sharing (Poline et al., 2012), more and more neuroimaging data are made available on dedicated platforms (*e.g.* OpenNeuro (Markiewicz et al., 2021) or NeuroVault (Gorgolewski et al., 2015)). Re-using shared data in studies would allow researchers to explore new research questions, with larger and more diverse datasets, while bypassing the difficulties associated with acquiring new data. Derived data

---
* Joint senior authorship

could also be combined instead of raw data through meta- and mega-analyses (Costafreda, 2009). This process also reduces privacy constraints and avoids costly re-computations.

In fMRI, due to the high flexibility of analytical pipelines (Carp, 2012), derived data shared on public databases often come from different pipelines. However, different pipelines can lead to different results (Botvinik-Nezer et al., 2020), and combining results from different pipelines in mega-analyses can lead to a higher risk of false positive findings (Rolland et al., 2022). To benefit from this large amount of derived data available, it is thus necessary to find a way to mitigate the effect of analytical variability and pipeline differences.

To mitigate the effect of domain shift, *i.e.* data submitted to different sources of variability (*e.g.* scanning devices), researchers usually perform data harmonization. Style transfer (Gatys et al., 2016), a technique that allows learning mappings between different domains, leverage generative models, such as Generative Adversarial Networks (GANs) (Goodfellow et al., 2014) and Diffusion Models (DMs) (Ho et al., 2020), for this task. Supervised frameworks (Isola et al., 2017; Saharia et al., 2022) can be trained to learn a mapping between data pairs from different domains. In contrast, unsupervised frameworks (Zhu et al., 2017; Sasaki et al., 2021; Liu et al., 2017) do not necessitate pairs of data in different domains for their training as they use constraints like cycle consistency (Zhu et al., 2017) or shared latent space assumption (Liu et al., 2017; Sasaki et al., 2021). These unsupervised frameworks provide a good opportunity to benefit from large unlabeled databases and learn complex mapping without ground truth. By conditioning on domain-specific features, unsupervised frameworks also extend to multi-domain (Choi et al., 2018, 2021; Ho and Salimans, 2021) to learn transfer between multiple domains in a single model.

In this work, we explore the ability of traditional style transfer frameworks to convert fMRI statistic maps between pipelines. Our goal is to find a solution to mitigate the effect of pipeline differences in fMRI statistic maps to build more valid mega-analyses and benefit from the large amount of derived data shared on public databases. To be useful in real practice, the framework should be trained on unpaired data and learn multi-domain transitions. To our knowledge, this application of style transfer to data conversion between different analysis pipelines is new, and off-the-shelf methods do not directly apply as these were not designed on the same type of data. Our contributions in this study are:

- We explore the potential of style transfer to harmonize fMRI statistic maps between pipelines. We hypothesize that pipelines can be seen as a style property, allowing their transfer across statistic maps while maintaining the intrinsic content.

- We evaluate three state-of-the-art style transfer frameworks based on GAN, namely Pix2Pix (Isola et al., 2017), CycleGAN (Zhu et al., 2017) and StarGAN (Choi et al., 2018), and adapt them to our 3-dimensional statistic maps. We also re-implement a state-of-the-art conditional DM (Ho and Salimans, 2021) and evaluate it for style transfer by conditioning the sampling on the source image.

- We explore different types of conditioning for DM: in particular using the latent space of an auxiliary classifier trained to distinguish pipelines, a task previously unexplored.

- We show that existing style transfer frameworks, particularly GAN, successfully perform pipeline-to-pipeline conversion of statistic maps, confirming our hypothesis and providing a new source of data to increase sample sizes in fMRI studies.

## Materials and Methods

### Dataset

We use group-level statistic maps from the *HCP multi-pipeline dataset*. More details can be found in (Germani et al., 2023). Briefly, this dataset is composed of subject-level (1,080 participants) and group-level (1,000 groups) statistic and contrast maps derived from raw data of the *Human Connectome Project Young Adult* S1200 release (Van Essen et al., 2013b). In this dataset, raw fMRI data for the motor task were analyzed with 24 different pipelines for the 5 contrasts: *right-hand*, *right-foot*, *left-hand*, *left-foot* and *tongue*. The pipelines used in this dataset vary in terms of software package, smoothing kernel Full-Width at Half-Maximum (FWHM), number of motion regressors, and derivatives of the Haemodynamic Response Function (HRF) included in the first-level analysis.

We explore in particular the statistic maps obtained with four different pipelines that differ in terms of software packages (SPM (Penny et al., 2011) or FSL (Jenkinson et al., 2012)) and the presence or absence of the derivatives of the HRF for the first-level analysis. We use all the available group-level statistic maps ($N = 1,000$) for each pipeline for the contrast *right-hand*. In the following, these pipelines are labeled with "<software>-<derivatives>", for instance "fsl-1" means the use of FSL software package with HRF derivatives.

The selected group-level statistic maps are resampled to a size of 48 x 56 x 48 and masked using the intersection mask of all groups. The voxel values are normalized between -1 and 1 for each statistic map using a min-max operation. The 1,000 groups are split into train and test with an 80/20 ratio and all models are trained and evaluated on the same sets. Further investigation about possible data leakage across groups is provided in Supplementary Figure 1 (Germani et al., 2024b).

### GAN frameworks

First, we assess the potential of GAN frameworks to convert statistic maps between pipelines. In particular, we evaluate the performance of Pix2Pix (Isola et al., 2017), CycleGAN (Zhu et al., 2017) and StarGAN (Choi et al., 2018). A detailed description of each framework is available in the corresponding papers. We provide a quick description of the main properties of these models in Table 1.

| Framework | Learning | Transition | Loss |
|---|---|---|---|
| Pix2Pix (Isola et al., 2017) | Supervised | One-to-one | Adversarial |
| | | | Reconstruction |
| CycleGAN (Zhu et al., 2017) | Unsupervised | One-to-one | Adversarial |
| | | | Cyclic |
| StarGAN (Choi et al., 2018) | Unsupervised | Multi-domain | Adversarial |
| | | | Cyclic |
| | | | Classification |

Table 1: Description of GAN frameworks

**Architecture and training**  We use the default architecture of these models, as described in their respective papers, and we only modify the 2-dimensional convolutions and batch

normalization layers to cope with our 3-dimensional statistic maps. These were implemented using PyTorch (Paszke et al., 2019) and each framework was trained for 200 epochs on 1 GPU NVIDIA Tesla V100.

### DM frameworks

Due to the promising performance of DM on natural images and medical imaging (Dhariwal and Nichol, 2021), we also assess the potential of DM frameworks. However, only a few DM frameworks have been developed for style transfer, and to our knowledge, they rely on paired data (Saharia et al., 2022) or learn only one-to-one transitions (Pan et al., 2023). Thus, to perform multi-domain transitions, we adapt an existing conditional DM to style transfer tasks. We use the framework from (Ho and Salimans, 2021), which generates images conditioned using a one-hot encoding of the class (*i.e.* class vector). We extend this model to conditioning based on the latent space of an auxiliary classifier, inspired from (Preechakul et al., 2022). Both are unsupervised frameworks, learning multi-domain transitions. A more detailed description of the original framework is available in (Ho and Salimans, 2021).

**Source content preservation**   To adapt our framework to style transfer, we first needed to find a solution to generate images that maintain the intrinsic properties of the source image. In (Saharia et al., 2022), authors concatenate the source image with random Gaussian noise to initialize the diffusion. Here, we propose to fix the initial state of the DM by directly using the forward diffusion process to generate a noisy version of the source image $X_t$. Then, the noisy source image is iteratively denoised using the predicted noise conditioned on the target domain and the reverse diffusion process.

**Classifier conditioning**   We also extend the model from (Ho and Salimans, 2021) with conditioning on the latent space of an auxiliary classifier. In (Ho and Salimans, 2021), the diffusion is conditioned using a one-hot encoding of the domain, limiting the diversity of generated samples. In (Preechakul et al., 2022), a semantic encoder is used to guide sampling. We extend this idea by conditioning the model on a latent feature vector extracted from a pre-trained CNN. This CNN was pre-trained to distinguish pipelines from the statistic maps. The features are extracted just before the classification layer, to get a good representation to distinguish images across pipelines.

**Multi-target images**   (Choi et al., 2021) showed that conditioning on multiple images generates images that share more coarse or fine features with the target ones depending on the number of selected images. Here, we aim to better represent the heterogeneity of the target domain. In practice, the whole set of images available in the target domain could be used. This is impractical for large datasets and might lead the model to focus on specific patterns of the target domain if these are over-represented in the dataset.

**Architecture and training**   The neural network used to predict the noise in the diffusion model follows a simple U-Net architecture (Ronneberger et al., 2015) with two downsampling and upsampling blocks with 3D convolution layers and skip connections. Hyperparameters are the following: $t = 500$ diffusion steps; linear noise schedule with variances in the range of $\beta_1 = 1e^-4$ and $\beta_t = 0.02$; batch size of 8 and learning rate of 1e-4. The model is implemented using PyTorch (Paszke et al., 2019) and trained for 200 epochs on 1 GPU NVIDIA Tesla V100.

The auxiliary classifier used to extract class conditional features contains five 3-dimensional convolution layers with 3-dimensional batch normalization and leaky rectified linear units (ReLU) activation functions, followed by a fully connected layer. The latent features correspond to $4,096$ flattened vectors injected as conditioning to the U-Net. It is trained for 150 epochs using a learning rate of 1e-4 and a batch size of 64 on 1 GPU NVIDIA Tesla V100.

**Evaluation metrics**

We used two types of metrics: Pearson's correlation coefficient and Mean Squared Percentage Error (MSPE) to study the adequacy of the generated images to the ground truth target, and Inception Score (Salimans et al., 2016) (IS) to explore the quality and diversity of the generated images. IS combines the confidence of the class predictions (*i.e.* each image's label distribution $p(Y|X)$) with the variety in the output of the model (*i.e.* the marginal label distribution for the whole set of images $P(Y)$). As an additional evaluation criterion, we used the auxiliary classifier to classify the generated images and verify if these images were correctly classified in the target pipeline class.

**Results**

| | IS | fsl-1 → spm-0 | spm-0 → fsl-1 | fsl-1 → spm-1 | fsl-1 → fsl-0 |
|---|---|---|---|---|---|
| | | ↑ Mean correlations (%) ± std. errors | | | |
| *Initial* | *3.69* | *78.2 ± 0.5* | *78.2 ± 0.5* | *82.8 ± 0.3* | *92.3 ± 0.5* |
| Pix2Pix | - | **91.4 ± 0.1** | **89.1 ± 0.2** | **90.1 ± 0.2** | **97.4 ± 0.1** |
| CycleGAN | - | 85.5 ± 0.3 | 67.1 ± 0.4 | 70.0 ± 0.5 | 71.2 ± 0.4 |
| StarGAN | 3.63 | 90.5 ± 0.4 | 86.8 ± 0.5 | 87.6 ± 0.5 | 91.5 ± 0.3 |
| One-hot | 3.66 | 83.9 ± 0.7 | 75.0 ± 0.9 | 78.8 ± 0.8 | 81.1 ± 0.6 |
| N=1 | 3.70 | 85.4 ± 0.6 | 77.4 ± 0.8 | 80.1 ± 0.8 | 82.8 ± 0.8 |
| N=10, ∞ | 3.86 | 86.1 ± 0.4 | 78.9 ± 0.6 | 81.5 ± 0.4 | 84.1 (0.6 |

| | fsl-1 → spm-0 | spm-0 → fsl-1 | fsl-1 → spm-1 | fsl-1 → fsl-0 |
|---|---|---|---|---|
| | ↓ Mean MSPE (%) ± std. errors | | | |
| *Initial* | *19.56 ± 1.95* | *19.56 ± 1.95* | *14.9 ± 1.75* | *1.04 ± 0.23* |
| Pix2pix | 0.9 ± 0.19 | **1.49 ± 0.45** | **0.87 ± 0.25** | **0.46 ± 0.11** |
| CycleGAN | 1.04 ± 0.39 | 1.98 ± 0.4 | 1.26 ± 0.38 | 2.53 ± 0.54 |
| StarGAN | **0.75 ± 0.23** | 3.11 ± 0.74 | 1.37 ± 0.23 | 1.38 ± 0.29 |
| One-hot | 2.9 ± 0.88 | 2.64 ± 0.63 | 3.2 ± 1.31 | 5.48 ± 1.16 |
| N=1 | 2.12 ± 0.61 | 3.11 ± 0.9 | 3.33 ± 0.67 | 4.04 ± 0.9 |
| N=10, ∞ | 1.4 ± 0.44 | 3.07 ± 0.88 | 2.04 ± 0.4 | 3.81 ± 0.79 |

Table 2: Performance of GAN and DM frameworks. IS means "Inception Score". Pearson's correlation (%) and Mean Squared Percentage Error (MSPE) are computed between generated and ground truth images and averaged across 20 images. *Initial* represents the metrics between the source image (before transfer) and the ground-truth target image. **Boldface marks the top model**.

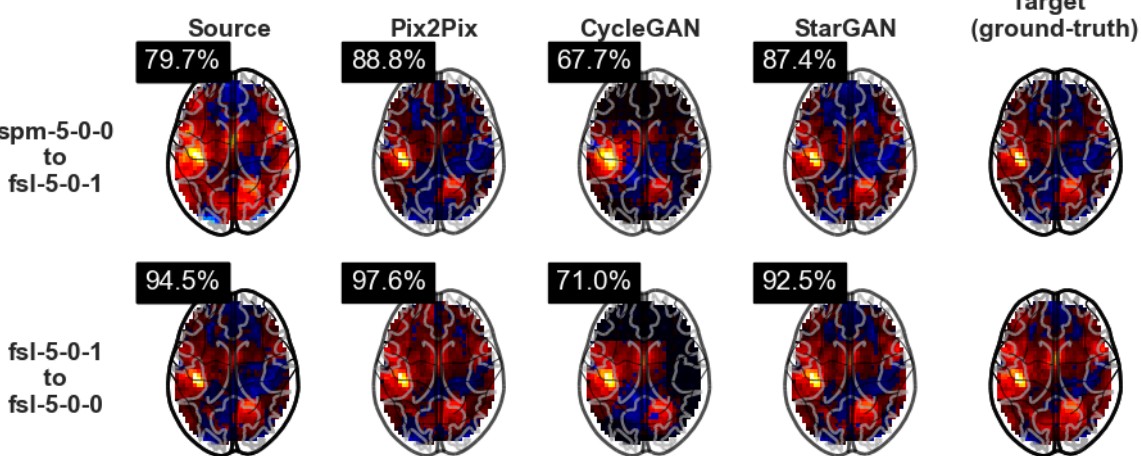

Figure 1: Generated images for two transfers and different competitors: Pix2Pix (Isola et al., 2017), CycleGAN (Zhu et al., 2017) and starGAN (Choi et al., 2018). Correlation with ground truth are indicated above each image (in percent).

## GAN frameworks

In Table 2, we show the performance of GAN frameworks for transfers between pipelines with: a different software and a different HRF (columns 1-2), a different software and the same HRF (columns 3) and, the same software and a different HRF (columns 4). Overall, using Pix2Pix (Isola et al., 2017) and StarGAN (Choi et al., 2018), the conversion of statistic maps between pipelines is successful, with increased correlations between target and generated maps compared to correlations between source and target, *e.g.* 91.4% for target-generated compared to 76.2% for source-target with Pix2Pix for conversion "fsl-1" to "spm-0". Regarding MSPE, we observe large error percentages between source and target maps (*initial*), especially for pipelines of different software packages. These large dissimilarities were masked in correlations, for instance, 14.9% of errors for "fsl-1" to "spm-1" and a correlation of 82.8%. The conversion seems more effective when observing MSPE than when observing correlations. The frameworks are trained using a reconstruction loss, based on a voxel-to-voxel comparison of errors, which is consistent with our observations on MSPE.

With both correlations and MSPE, we can point out the large superiority of the supervised Pix2Pix framework compared to unsupervised GAN alternatives. By benefiting from paired data, Pix2Pix generates images closer to the target image than the source image for all transfers. Correlations between target and generated images are close to 90%, which is nearly perfect. For comparison, in the NARPS study (Botvinik-Nezer et al., 2020), highly correlated statistic maps had a correlation coefficient between 44% and 85%. On the other hand, the CycleGAN (Zhu et al., 2017) framework gives surprising results, relatively low compared to other frameworks, especially in correlation measures. While it makes use of a cyclic loss, similarly to StarGAN (Choi et al., 2018), it only learns transfers between two

domains. We can suppose that StarGAN leverages the multiple source domains and benefits from the additional classification loss, leading to higher performance than CycleGAN.

In Figure 1, we illustrate two transfers: (first row) between pipelines with different software packages and different HRF (spm-0 to fsl-1) and (second row) between pipelines with the same software package and different HRF (fsl-1 to fsl-0). We randomly selected a statistic map of the source pipeline and generated the corresponding converted map in the target pipeline. We also display the ground-truth map in the target pipeline. Maps generated using Pix2Pix and StarGAN are close to the ground truth, with highly similar activation patterns, consistent with the similarity metrics.

### DM frameworks

We also show the performance of the DM frameworks for the same four transfers in Table 2. Several frameworks are compared: one-hot conditioning from (Ho and Salimans, 2021), auxiliary classifier-conditioning with $N = 1$ target image selected randomly, inspired from (Preechakul et al., 2022), and auxiliary classifier-conditioning with $N = 10$ target images selected randomly (named $N = 10, \infty$ in the Table).

The conversion between pipelines seems less successful than with the GAN frameworks, especially when observing correlation metrics. While all frameworks succeed in changing the class identified for the generated maps by a pipeline classifier from the source to the target domain, the success of the conversion in terms of similarity to the ground truth is variable. For instance, all DM frameworks succeed for the transfer "fsl-1" to "spm-0", but none is successful for the transfer "fsl-1" to "fsl-0". These low performance could be explained by the difficulty of the models to learn differences between pipelines that provide similar results (*i.e.* whose results display very similar activation patterns).

When observing MSPE values, generated statistic maps seem close to the ground-truth targets, with percentage errors between 1.4 and 3.8% for the best DM framework (N=10, $\infty$). While these values remain higher than those observed with GAN frameworks, MPSE between generated and target maps are still lower than initial MPSE (around 15-20% for pipelines of different software packages). This is consistent with our observations on MPSE values obtained with GAN frameworks, showing higher similarity between generated and target maps than correlations. These differences could be related to the different scaling strategies of the two software packages. Low MSPE values highlight the success in the conversions of activation patterns, but also in the adaptation to the software package value scale. On the other hand, correlation metrics mostly focus on differences in terms of distributions of values and are independent of the scales of values.

Using a DM with auxiliary classifier conditioning and multiple target images ($N = 10, \infty$) improves the performance compared to the alternative frameworks. Both the quality and diversity of images are increased ($IS = 3.86$). In terms of similarity to the ground truth, this framework also outperforms other DM models by up to 4% in correlations between ground-truth and generated image compared to (Ho and Salimans, 2021) for the transfer "spm-0" to "fsl-1" and up to 3% for "fsl-1" to "spm-0".

The first row of Figure 2 illustrates a transfer between pipelines with different software packages and different HRF ("spm-0" to "fsl-1"). The second row shows a transfer between pipelines with the same software package and different HRF ("fsl-1" to "fsl-0"). DM with

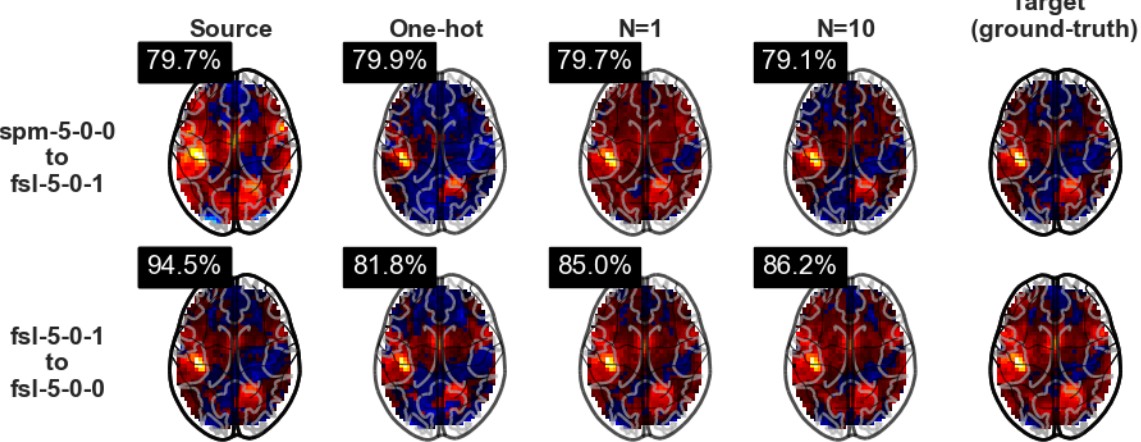

Figure 2: Generated images for two transfer and different competitors: conditioning with one-hot encoding (Ho and Salimans, 2021), with a classifier-conditioning N=1 and N=10 target images randomly selected. Correlations with ground truth are indicated above generated and source images (in percent).

multiple target images generates statistic maps close to the ground truth for both transfers, representing the intrinsic properties of the map while modifying its extrinsic properties to the target domain. Using the one-hot encoding, generated statistic maps seem far from the ground truth, failing to represent the full characteristics of the target domain.

**Usefulness in real practice**

While better performances are achieved using Pix2Pix(Isola et al., 2017), StarGAN exhibits better usefulness in real practice thanks to its unpaired training process and ability to learn multi-domain transitions. The performance of DM frameworks remains notably inferior to the ones obtained with StarGAN (Choi et al., 2018). This superiority can be explained by the differences between frameworks: GAN methods use adversarial training and StarGAN improves this by using a classifier loss and a cyclic-reconstruction loss. Moreover, the sampling process of GAN relies on the source image directly and does not require setting an initial state, which might facilitate the source content preservation. However, we can note that Inception Scores (IS) obtained with DM frameworks are better than the ones obtained with StarGAN, which indicates that images generated by DM frameworks are more diverse. This observation is consistent with the literature (Dhariwal and Nichol, 2021) and the sampling process of DM frameworks which includes randomness. Thus, we conclude on the superiority of the StarGAN framework for our task and potential downstream analyses.

Overall, for both GAN and DM frameworks, some transfers seem easier to learn than others. We hypothesized in the previous section that these variations in the success of transfers could be related to the similarity of initial maps, *i.e.* transfer between highly similar pipelines such as "fsl-1" to "fsl-0" are harder to learn. We can also see that for the same two pipelines, the conversion seems easier to learn in one direction: for instance,

between "fsl-1" and "spm-0", generated maps are closer to the ground-truth targets when transferring from "fsl-1" to "spm-0" (correlations of 90% with StarGAN) than when transferring from "spm-0" to "fsl-1" (correlations of 86.8% with StarGAN). In Figure 1, statistic maps from "spm-0" seem to contain more activated voxels (red values) than the ones from "fsl-1". We can suppose that for the task of transferring from a statistic map with few activated voxels to a map with more activated voxels, the mapping is easier to learn, maybe due to the similarity of this task to dilations of already activated areas. On the opposite, when converting between pipelines with larger activation areas to sharper ones, the erosion task seems more difficult to learn. In practice, when they have the choice, we could thus recommend to researchers who would like to use style transfer to harmonize statistic maps to favor the conversion of maps to pipelines with larger activation areas.

## Conclusion

In this study, we explored the potential of style transfer frameworks for converting fMRI statistic maps between different pipelines. We show that GAN frameworks are particularly effective for this task. In particular, the StarGAN framework could be useful in real practice, thanks to its unpaired training process and ability to learn multi-domain transitions. Researchers could easily train and apply this framework on derived data (*e.g.* statistic maps from multiple unknown pipelines) shared on public databases to harmonize them and combine them in larger meta- or mega-analyses. By maintaining the intrinsic properties of brain activity while changing the style of the image, these newly generated maps could be a source of data augmentation to build more valid fMRI statistical studies. Future works will focus on measuring the validity of statistical studies performed using converted statistic maps. In particular, we would like to compare the results of mega-analyses computed with harmonized data instead of original data from different pipelines. We expect that, with this conversion step, statistical studies combining data from different pipelines would have fewer invalid results (Germani et al., 2024c). This would allow researchers to re-use derived data in larger statistical analyses, potentially answering new research questions without the need to acquire new data or risking invalid results due to the use of different pipelines.

## Acknowledgments

This work was supported by Region Bretagne (ARED MAPIS) and ANR project ANR-20-THIA-0018). Data were provided by the Human Connectome Project, WU-Minn Consortium (Principal Investigators: David Van Essen and Kamil Ugurbil; 1U54MH091657) funded by the 16 NIH Institutes and Centers that support the NIH Blueprint for Neuroscience Research; and by the McDonnell Center for Systems Neuroscience at Washington University.

## Data statement

This study was performed using derived data from the HCP Young Adult (Van Essen et al., 2013b), publicly available at ConnectomeDB. Data usage requires registration and agreement to the HCP Young Adult Open Access Data Use Terms available at: (Van Essen et al., 2013a).

The HCP multi-pipeline dataset (Germani et al., 2023) is publicly available on Public-nEUro (at Rigshopitalet, 2023): https://publicneuro-catalogue.netlify.app/dataset/PN000003%20HCP%20multipipelines/V1.

## Ethics

This study was performed using derived data from the HCP Young Adult (Van Essen et al., 2013b). No experimental activity involving the human participants was made by the authors. Only shared data were used.

Written informed consent was obtained from participants and the original study was approved by the Washington University Institutional Review Board.

We agreed to the HCP Young Adult Open Access Data Use Terms available at: (Van Essen et al., 2013a).

## Code and data availability

All the scripts used to perform the study (models training, testing and performance evaluation) are available on Software Heritage: swh:1:snp:b0b52aa88bef8f4411bdd7e00a2d71715d7830bb (Germani et al., 2024a).

Derived data such as pre-trained models and computed metrics are available on Zenodo (Germani et al., 2024b).

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

## Supplementary materials

### Generalizability study

Ideally, researchers should be able to re-use a model trained to convert statistic maps of a task for another one. Thus, we test the generalizability of the frameworks trained to convert statistic maps from a particular task to statistic maps of other tasks. Here, we explore the generalizability of GAN frameworks trained on maps of *right-hand* to maps of *right-foot* and *left-hand*. Due to its potential for real practice applications, we particularly focused on the generalizability of StarGAN, trained on unpaired data for multi-domain transitions.

| | fsl-1 → spm-0 | spm-0 → fsl-1 | fsl-1 → spm-1 | fsl-1 → fsl-0 |
|---|---|---|---|---|
| **Converting *right-foot* statistic maps with:** | | | | |
| *Initial* | *86.2* | *86.3* | *85.8* | *95.9* |
| Trained on *right-foot* | 90.5 | 88.8 | 88.9 | 93.1 |
| Trained on *right-hand* | 71.2 | 71.0 | 63.0 | 82.2 |
| **Converting *left-hand* statistic maps with:** | | | | |
| *Initial* | *82.3* | *82.3* | *85.9* | *92.9* |
| Trained on *left-hand* | 88.4 | 85.4 | 86.8 | 85.8 |
| Trained on *right-hand* | 73.2 | 74.5 | 69.5 | 75.2 |

**Supplementary Table 1** - Robustness to distribution shifts (*i.e.* trained and evaluated with statistic maps from different tasks) of StarGAN. Pearson's correlation (%) is computed between generated and ground truth and averaged across 20 images per transfer. Initial represents the metrics between the source image (before transfer) and the target image.

Supplementary Table 1 shows the performance of the StarGAN framework trained on *right-hand* statistic maps when applying it to statistic maps of *right-foot* and *left-hand*. We also show the performance of StarGAN trained directly on *right-foot* and *left-hand* for comparison. Our results show that the StarGAN framework does not behave positively when applied to a task different from the task used in the training data: the generated images seem further from the target images (*e.g.* 71.2% for the first transfer between "fsl-1" to "spm-0") than the source image was from the target (*e.g.* 86.2% in this case). We can see a large performance drop between the framework trained on statistic maps from the task compared to the one trained on maps from another task. Similar observations can be made for generalizability to closer tasks (here, frameworks trained on *right-hand* evaluated on *left-hand*). These results make us suppose that the mapping from one pipeline to another is different between tasks and thus, if the same model can be used for different subjects on the same task, different models should be trained to transfer statistic maps from tasks unseen during training.

**Supplementary figures**

| Pipelines | Layer 1 | Layer 2 | Layer 3 | Layer 4 |
|---|---|---|---|---|
| Same software, different parameters | | | | |
| fsl-5-0-0 / fsl-5-0-1 | 86.5 | 91.4 | 95.4 | 99.2 |
| spm-5-0-0 / spm-5-0-1 | 86.5 | 90.9 | 94.2 | 98.4 |
| Same parameters, different software | | | | |
| fsl-5-0-0 spm-5-0-0 | 88.8 | 88.2 | 93.6 | 98.2 |
| fsl-5-0-1 spm-5-0-1 | 84.8 | 85.8 | 92.4 | 98.0 |
| Different software, different parameters | | | | |
| fsl-5-0-0 spm-5-0-1 | 74.5 | 81.0 | 88.7 | 97.1 |
| fsl-5-0-1 spm-5-0-0 | 74.8 | 77.7 | 88.2 | 97.3 |

**Supplementary Table S1.** Mean correlations between feature maps learned at each layer for each pair of pipelines. Features are close for pipelines sharing the same software at Layer 4, which might explain the difficulty of relying on these features to perform a transfer.

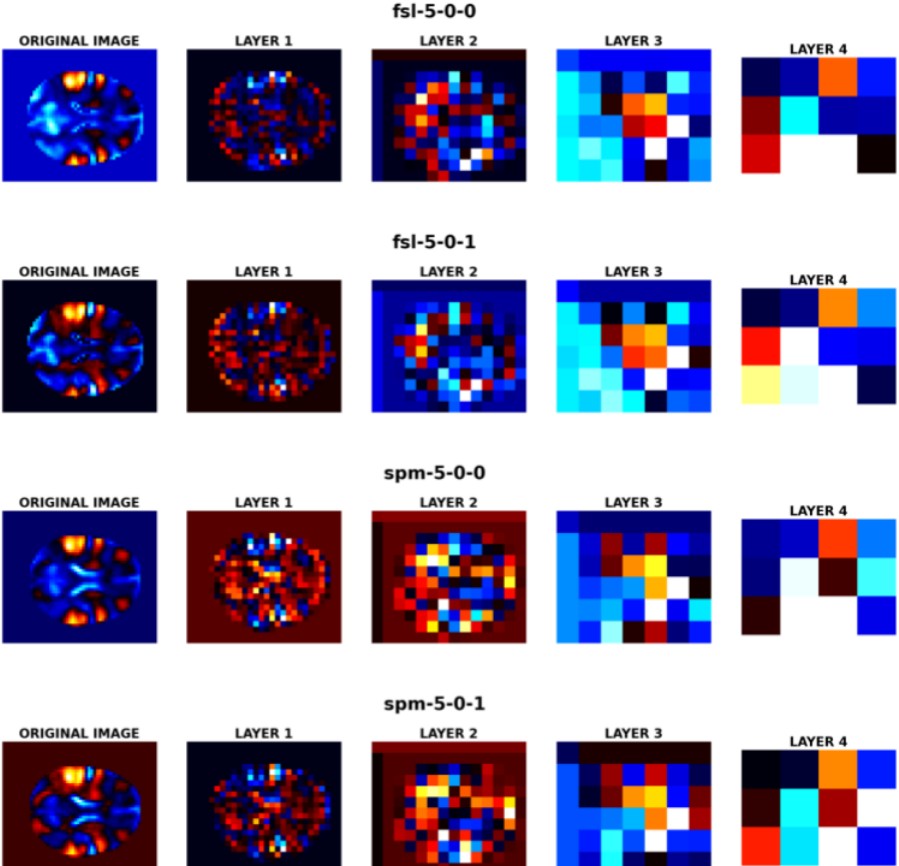

**Supplementary Figure S1** - Original mean statistic maps (column 1) and mean feature maps across groups learned by the pipeline classifier for the first 4 convolutional layers for the different classes. Pipelines with the same software show similar feature maps at Layer 2 and 3.

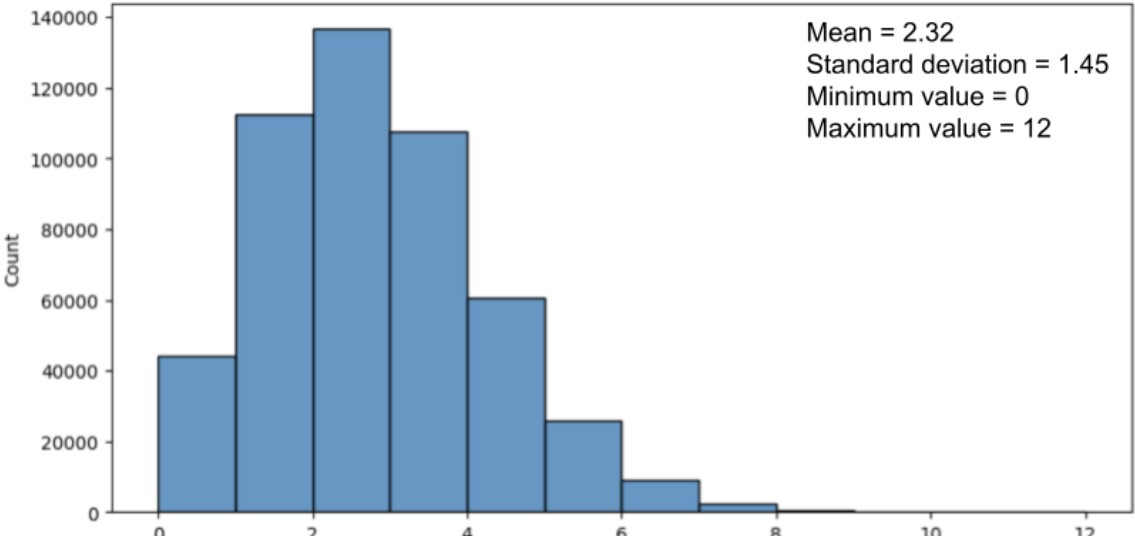

**Supplementary Figure S2** - Histogram of the number of shared participants between two groups for each pair of groups across the whole dataset. While shared participants between groups can impact our results by slightly over-estimating our performance, the impact is likely to be low due to the small number of shared participants (2.3 on average). In addition, this has no impact the conclusions of our study on the comparison of performance between different models as all models are trained and evaluated on the same sets of groups.

