# OpenReview forum: "Mitigating analytical variability in fMRI with style transfer"
_MIDL.io/2025/Conference — MIDL 2025 Poster_

### Official Review · Reviewer_rfNn · 2025-02-17

**Confidence:** 3
**Preliminary Rating:** 3
**Recommendation:** Poster
**Final Rating:** 3

**Summary:**

Many different pipelines are being used in fMRI that make comparisons across different studies difficult. In this work, the authors propose using unsupervised image to image translation methods to transform the statistic maps form one pipeline to another. To this end, they use a Pix2Pix, CycleGAN, StarGAN and a conditional denoising diffusion model.

**Strengths:**

The paper motivates the work well, as the fMRI pipelines are reportedly quite heterogeneous. Being able to either homogenize them or to translate between the different pipelines would simplify including multiple datasets from multiple sources. Tackling this problem is an important step in being able to reutilise already recorded data, and in the context of fMRI this is especially important due to the high cost in acquiring said data. The authors also include multiple different approaches including GAN-based and denoising diffusion models. They also show the performance across multiple different types of pipelines from different software packages. The authors use a public dataset and shared the code used for the experiments in public repositories.

**Weaknesses:**

The results are not very strong. Table 2 and Table 3 show that the correlations between the source and target pipeline are already relatively high (0.792 - 0.923), and the proposed models with the exception of the only supervised model “Pix2Pix” do not seem to contribute any or only a weak improvement. While the authors deem their proposed models “successful”, it is not clear what that means, especially with respect to the unsupervised models. The statement could be made much stronger by showing how the translated data improves e.g. the performance on downstream tasks. I understand that it is interesting to consider unsupervised models, but as they do not seem to provide any improvement over just using the source maps as they are. In that light I think it would have been preferable to instead focus more on the method that did bring some improvement, and maybe evaluate it more extensively. Finally, the method was trained and evaluated on a dataset derived from HCP-YoungAdult. To strengthen the argument of enabling and combining multiple data sources, it would have been worthwhile to include some other datasets in the analysis.

**Detailed Comments:**

There are some minor things/typos that caught my eye, and I would like to ask the authors to consider if they need to be addressed for a final version:

- In section “Architecture and Training” you report $\beta_1 = 10^4$, I assume this should say $10^{-4}$ instead?

    The overlay of the symbolic brain drawing in Figure 1 and 2 distract from looking at the underlying plots, I would propose removing them.

- For the ease of comparison, I would propose merging Table 2 and 3 as they report the same measures just for a different set of models.

- In Table 2 the MSE for StarGAN in (spm-0->fsl-1) has a digit less than the other reported values.

- The MSE that is reported is hard to interpret given that the range of values that the compared image sets can attain is not reported. What scaling does MSE reported in the two tables refer to? If it is computed on the [-1,1]-normalized images, that seems to imply that mean that every map was scaled a little bit differently, and therefore does not contribute in the same way to the MSE (e.g. small errors on images that only have values in the range [90,120] would weight just as much as larger errors on images with values in the range [0, 255]). Alternatively I would propose the PSNR which directly measures the error relative to the original data range.

**Justification Of The Final Rating:**

I'd like to thank the authors for answering the questions in their rebuttal. I am still not entirely convinced of the choice of methods and their assessment. I apprciate that the authors clarified some paragraphs in their manuscript and that they addressed some of the issues that were raised.

**Justification Of The Preliminary Rating:**

The motivation of the work is clearly stated, however I am not convinced by the performance of the proposed methods, as well as the assessment of the evaluation of those methods. The paper could be made stronger in my view by better motivating the various choices criticized above.

**Questions To Address In The Rebuttal:**

1. All proposed models seem to perform worse on the (spm-0 > fsl-1)-task compared to the other direction. Do you have any intuition or explanation why this is the case? Do the fsl-1 images inherently contain more information?

2. In the results from the GAN frameworks it is mentioned that a correlation of 0.9 is considered “nearly perfect”. Is this number motivated by some downstream task or other criteria?

3. In the discussion you primarily refer to StarGAN. Can you elaborate why you in the end consider this the most interesting method for this problem?

4. What downstream analysis do you think would profit most from the proposed models? Are there any specific directions of research you intend to use this work for?

**Special Issue:**

No

---

> ### Author Response · Authors · 2025-03-06
>
> We thank the reviewer for their comment, and answer each point in the following:
> 1. All proposed models seem to perform worse on the (spm-0 > fsl-1)-task compared to the other direction. Do you have any intuition or explanation why this is the case? Do the fsl-1 images inherently contain more information?
> All proposed models seem to perform worse on the (spm-0 -> fsl-1) task compared to the inverse (fsl-1 -> spm-0), e.g. for the same source correlation, converted statistic maps are closer to the target in the second task (correlations generated-target around 0.90 vs. 0.85). In Figure 1, we can see an example of statistic maps from pipeline spm-0 (source) and fsl-1 (target). Statistic maps from spm-0 seem to contain more activated voxels (red) than the ones from fsl-1. We can hypothesize that for the task of transferring from a statistic map with few activated voxels to a map with more activated voxels, the models perform better, as they can see this task as a dilation of already activated areas. On the opposite, when transferring from larger activation areas to sharper ones, the erosion task seems to be harder to learn. We added a section in the results to better develop usefulness in real practice and highlighted this point, in particular as this could help researchers choose the direction of the conversion to use when they can.
>
> 2. In the results from the GAN frameworks it is mentioned that a correlation of 0.9 is considered “nearly perfect”. Is this number motivated by some downstream task or other criteria?
> In the literature, several studies computed correlations between statistic maps of different pipelines. Our study is performed on pipelines with few differences (one or two parameters diverging, source-target correlations are already high) compared to the variability between pipelines that can be found on public databases. This threshold of 0.9 to consider correlation as nearly perfect was inspired by previous studies, e.g. NARPS, where statistic maps of different pipelines were considered as strongly correlated if the correlation was in the range of 0.44–0.85. Here, due to the initially high correlation, we used a 0.9 threshold. We added a sentence in the manuscript to compare with NARPS.
>
> 3. In the discussion you primarily refer to StarGAN. Can you elaborate why you in the end consider this the most interesting method for this problem?
> As stated in the introduction, to be useful in real practice and allow researchers to re-use derived data by converting them between pipelines, style transfer frameworks should be able to be trained on unpaired data (i.e. no match between source and target). While pix2pix showed highly interesting results, it requires pairs of images for training, which are rarely available, particularly in public databases. Thus, an unsupervised framework such as StarGAN would be more easy to use in practice. We added a section (see Usefulness in real practice) to better explain this choice.
>
> 4. What downstream analysis do you think would profit most from the proposed models? Are there any specific directions of research you intend to use this work for?
> In another study (Germani et al., 2024), we showed that mega-analyses with subject-level data processed with different pipelines were leading to invalid results (false positive rates > 0.05). In this study, we try to convert statistic maps at the group level between pipelines, providing a first evaluation of the potential of style transfer techniques to convert fMRI derived data between pipelines. In the future, we would like to adapt such frameworks to subject-level data, to build more valid mega-analyses. We could test the quality of converted data using downstream analysis such as comparing the results of one-sample or two-sample t-tests with data from a single pipeline, data from different pipelines, and data from a single pipeline + data from different pipelines converted to the target. In future work, to go even further, our project will focus on the validity of group-level meta-analyses with converted data. We added a few sentences to the conclusion to explain it better.
>
> Germani, E. et al., “On the validity of fMRI studies with subject-level data processed through different pipelines”. 2024. Accepted for publication in Imaging Neuroscience.

---

> > ### Comment · Reviewer_rfNn · 2025-03-06
> >
> > I'd like to thank the authors for answering the questions. In the updated manuscript the sections "Usefulness in practice" and the Conclusion are now definitely clearer to understand in my opinion.

---

### Official Review · Reviewer_vovD · 2025-02-18

**Confidence:** 2
**Preliminary Rating:** 4
**Recommendation:** Poster
**Final Rating:** 4

**Summary:**

The paper utilizes style transfer mechanisms for conversion of fMRI statisc maps between pipelines, thus, attempting to reduce the effect of pipeline differences for fMRI mega-analyses. The authors claim that this could be useful in enabling the usd of large data available through public repositories. Essentially, it is an application of domain-to-domain transformation.

**Strengths:**

Experiments are thoroughly reported.
Multiple architectures are explored and utilised.
Pearson’s correlation coefficient and Mean Squared Error are calculated to evaluate the performance.
References to relevant literature are provided.
The work shows some interesting experiments on transfer of fMRI maps, and could be of interest to the audience.

**Weaknesses:**

No new model is suggested.
A comparative study is performed on three style transfer models (two GANs and one diffusion model). And the observation is that diffusion model achieves better inception score. Not a surprise.

**Detailed Comments:**

no more comments.

**Justification Of The Final Rating:**

I have read the authors' responses and I believe they have nicely addressed my concerns and reported some addtional comparison too. Based on these responses, I have changed my voting for this and would be happy to see it accepted.

**Justification Of The Preliminary Rating:**

I am convinced that the authors have reported some thorough experiments and that the work will present a good case at the MIDL conference. While there are some limitations in terms of novel development, the application is still interesting.

**Questions To Address In The Rebuttal:**

What new knowledge is gained from this experiment?

**Special Issue:**

No

---

> ### Author Response · Authors · 2025-03-06
>
> We thank the reviewer for their comment, and answer each point in the following:
> 1. What new knowledge is gained from this experiment?
> In this study, we explore the ability of existing style transfer frameworks to convert fMRI statistic maps between pipelines. While such frameworks are often used to convert medical imaging data between different sources of variability (e.g. different scanning devices), to our knowledge, it has never been used to convert statistic maps across pipelines. Our experiments show that pipelines can be considered as a style component of statistic maps and transferred across subjects and groups. This new knowledge provides an opportunity to re-use derived data (here, statistic maps) shared in public databases: by using style transfer frameworks to convert data between pipelines, researchers could build larger and more diverse studies while taking into account analytical variability and pipeline differences. While we did not explicitly assess the validity of mega-analyses using converted statistic maps, our similarity metrics and classification pipelines provide strong evidence that conversion is successful, and the validity of downstream tasks will be explored in future works. We made changes in the introduction, results, and conclusion sections to better highlight the rationale and usefulness of our experiments, and how our results will determine the future works.

---

### Official Review · Reviewer_WkuD · 2025-02-21

**Confidence:** 3
**Preliminary Rating:** 3
**Recommendation:** Poster
**Final Rating:** 4

**Summary:**

The authors proposed a new approach to mitigate fMRI pipeline differences by performing style transfer using various generative models. They implemented CycleGAN, Pix2Pix, and StarGAN for performing the style transfer on the HCP multipipeline dataset. Their goal is to transfer the 3D statistics maps from one pipeline result to the other. For evaluation, they used Pearson's correlation and average MSE, and found that Pix2Pix is the best-performing model.

**Strengths:**

1. The authors reimplemented the state-of-the-art generative models (Pix2Pix, CycleGAN, StarGAN) to adapt to the 3D inputs and compared them thoroughly on different pipeline derivatives.
2. The idea of using style transfer to mitigate fMRI pipeline variability is innovative and has strong potential for data augmentation in fMRI analysis and deep learning studies.
3. The authors also explored the latent space of the generative models and utilized a classification model to condition.

**Weaknesses:**

1. In the generalizability study, authors only presented the results for using StarGAN, while Pix2Pix was shown to be the best among all generative models. Adding Pix2Pix results here might also be helpful.
2. The authors did not experiment with a particular use case with these generative models.

**Detailed Comments:**

1. Supplementary Figure S1: text might be too small.
2. Figure 2: The statistics map may not align with the brain atlas well.
3. The mean MSE and standard deviation errors seem very small, and therefore, it might be difficult to perceive the relative magnitude of the value. Are there alternative metrics, such as mean percentage error?

**Justification Of The Final Rating:**

The authors have addressed questions from my initial review and explained more in detail the feasibility of using a style transfer framework on fMRI processing pipelines. The authors haven't analyzed the converted data showcasing the actual use case of this methodology, but indicated their plans in their future work.

**Justification Of The Preliminary Rating:**

The authors presented the comparison of the performance of the generative models and determined which one is the best in the style transfering task. However, this paper does not clearly show how these generated data could be used. For example, a comparison of a task classifier trained on the generated dataset or the real dataset. It would be beneficial if it could show that the classifier actually could use this as a data augmentation method.

**Questions To Address In The Rebuttal:**

1. What are the mean percentage error for each of the model performances?
2. The Pix2Pix has the best performance, but why choose starGAN instead of Pix2Pix for the generalization study?

---

> ### Author Response · Authors · 2025-03-06
>
> We thank the reviewer for their comment, and answer each point in the following:
> 1. What are the mean percentage error for each of the model performances?
> We thank the reviewer for suggesting mean percentage as a metric and concur that it is better suited than MSE to evaluate model performance. We modified Tables 2 and 3 with MSPE values (Mean Squared Percentage Error). Results show similar trends as for correlations and Mean Squared Error. However, we can see that MSPEs between source and target images are higher, highlighting original voxel-to-voxel dissimilarities, probably caused by the different scaling strategies of software packages, that are masked for correlations. This metric shows greater similarity between generated and target maps, for all models and transfers, compared to source-target comparison. We modified the results section to 1/ merge Tables 2 and 3 and 2/ comment on the MSPE results that give us more insight into the success of the conversion in terms of the scale of voxel values.
>
> 2. The Pix2Pix has the best performance, but why choose starGAN instead of Pix2Pix for the generalization study?
> As stated in the introduction, to be useful in real practice and allow researchers to re-use derived data by converting them between pipelines, style transfer frameworks should be able to be trained on unpaired data (i.e. no match between source and target). This generalization study aimed to further explore the potential of style transfer by assessing if a framework trained to convert statistic maps between pipelines in a given study context (here, a particular task) could be applied directly to convert statistic maps from another task, without retraining or fine-tuning. While pix2pix showed highly interesting results, it requires pairs of images for training, which are rarely available in practice, particularly in public databases. We modified the results section to include a section on the comparison of frameworks in terms of usefulness for real practice. We also added a few sentences in the conclusion and at the beginning of the Appendix to better explain our choice.
>
> We also thank the reviewer for the detailed comments, we have modified the supplementary figure to increase caption size. We also modified MSE to MPSE, in accordance to other reviewers comments.

---

### Comment · Area_Chair_Av53 · 2025-03-04
**rebuttal process**

Dear Authors,

I encourage you to actively participate in the rebuttal process, with a hard deadline on March 7 23:59 AoE.

You can leave official comments on OpenReview to address each reviewer's points separately, and upload a revised manuscript using the “rebuttal” function in OpenReview. The revision can include any additional details, experiments, or images that might be required in the paper within the page limit (max 9 excluding references, acknowledgements, and appendices), as well as other supporting documents in the rebuttal stage. Any changes must be highlighted in the revised manuscript.

Thank you

---

### Author Rebuttal · Authors · 2025-03-06

**Rebuttal:**

We thank the reviewers for their valuable feedback. We made the following changes in the revised manuscript.

*Novelty and potential use cases*
We highlighted the novelty of our work and its potential in the introduction, conclusion, and results sections. Our study explores style transfer frameworks to convert fMRI statistic maps between pipelines. While style transfer has already been applied in medical imaging, its use for harmonizing fMRI-derived data across pipelines remains unexplored. Our findings demonstrate that pipelines can be treated as a style component, allowing conversion across statistic maps. This opens new possibilities for reusing publicly available derived data, enabling larger and more diverse meta- and mega-analyses. We emphasize StarGAN’s potential as it can be trained on unpaired data, making it more practical than Pix2Pix, which requires paired images. A new subsection has been added to discuss these real-world applications.

*Results*
We have revised the results section by merging Tables 2 and 3 and replacing MSE with MSPE (Mean Squared Percentage Error), which provides better interpretability of voxel-scale transformations. MSPE highlights the inherent dissimilarities between source and target images due to different scaling strategies used by software packages, insights that were previously masked by correlation metrics. This metric further confirms that generated images align more closely with target images than source images, reinforcing the effectiveness of our approach. Additionally, we address the observed asymmetry in conversion performance, hypothesizing that transitioning from small to large activation areas (dilation) is easier than the reverse (erosion), which may guide future usage of this technique for converting statistic maps between pipelines.

*Future works*
We explain in more detail our long-term goal, i.e. to extend style transfer techniques to subject-level data, allowing for more valid mega-analyses. Prior work (Germani et al., 2024) has shown that combining subject-level data processed with different pipelines can lead to false positive findings. By converting data at the group level in this study, we take a first step toward harmonizing fMRI-derived data across pipelines. Future work will evaluate the impact of style-transferred data on statistical tests and explore the feasibility of using converted data in mega- and meta-analyses. The conclusion has been refined to reflect these future research directions.

**Supporting Material:**

/attachment/b5beddfccf067bf784849ac0453042b1923f6b1a.pdf

---

### Comment · Area_Chair_Av53 · 2025-03-10
**rebuttal process**

Dear Reviewers,

Please look at the authors' reply to your initial review. I encourage you to engage in discussion and consider the author replies, and update your rating and assessment (if justified) until the 14th of March.

Based on the initial review, the rebuttal discussion, and your final grades, I will write meta-reviews and propose a decision to the PC.

---

### Meta-Review · Area_Chair_Av53 · 2025-03-19

**Recommendation:** Accept (Poster)
**Confidence:** 4

**Metareview:**

The authors propose an innovative application of deep learning, bridging with SPM-style maps (such as obtained from mass-univariate statistical analysis) to transfer pipeline effects. All reviewers had a positive view of the work.

On the plus side, the reviewers noted the usefulness of the approach enabling wider reuse of data, thorough experiments including multiple pipelines and image-to-image models, and open science quality.

As weaknesses, the reviewers noted the lack of downstream use of the syle-transfered maps, relatively modest performance improvements (in correlation) after image-to-image translation, and use of a single dataset.

The authors engaged with the rebuttal process, changed the evaluation metric based on feedback, and improved their paper.